# Study protocol for cholera vaccination as a model to measure the inflammatory response in the gut: A case of modulation with a *Lactobacillus plantarum* K8 lysate

**Min Young Park[1]☯, Soo-yeon Park[2]☯, Anita Hartog[3], Els van Hoffen[3], Alwine Kardinaal[3], Joohee Kim[4], Hee Jung Choi[5], Oran Kwon[1,6]\*, Ji Yeon Kim[2,7]\***

**1** Department of Nutritional Science and Food Management, Ewha Womans University, Seoul, Republic of Korea, **2** Department of Food Science and Technology, Seoul National University of Science and Technology, Seoul, Republic of Korea, **3** Department of Health, NIZO, Ede, The Netherlands, **4** BiofoodCRO Co., Ltd., Seoul, Republic of Korea, **5** Division of Infectious Diseases, Office of Infection Control, Ewha Woman's University Medical Center, Seoul, Republic of Korea, **6** System Health & Engineering Major in Graduate School, Ewha Womans University, Seoul, Republic of Korea, **7** Department of Nutritional Science and Food Management, Graduate Program in System Health Science and Engineering, Ewha Womans University, Seoul, Republic of Korea

☯ These authors contributed equally to this work.
\* orank@ewha.ac.kr (OK); jiyeonk@seoultech.ac.kr (JYK)

**Data Availability Statement:** Deidentified research data will be made publicly available when the study is completed and published.

## Abstract

It is crucial for human health that the immune system of the gastrointestinal tract works effectively. Dietary modulation is one of the factors that regulate the immune response in the gut. This study aims to develop a safe human challenge model to study gastrointestinal inflammation and immune function. This study focuses on evaluating gut stimulation induced by the oral cholera vaccine in healthy people. In addition, this paper describes the study design for assessing the efficacy and safety of a probiotic lysate, identifying whether functional ingredients in food can modulate inflammatory response induced by oral cholera vaccine. Forty-six males aged 20 to 50 with healthy bowel habits will be randomly allocated to the placebo or intervention group. Participants will consume 1 capsule of probiotic lysate or placebo twice daily for 6 weeks, take oral cholera vaccines on visit 2 (day 15) and visit 5 (day 29). The level of fecal calprotectin, a marker of gut inflammation, will be the primary outcome. The changes of cholera toxin-specific antibody levels and local/systemic inflammatory responses will be evaluated in blood. The purpose of this study is to evaluate gut stimulation of the oral cholera vaccine and investigate the effect of a probiotic lysate on improving the mild inflammatory response induced by the vaccine or supporting the immune response in healthy subjects.

**Trial registration**: \* This trial is registered in the International Clinical Trials Registry Platform of WHO (ICTRP, registration number: KCT0002589).

**Funding:** This work was primarily supported by the "Bio&Medical Technology Development Program" of the Natl. Research Foundation (NRF) funded by the Ministry of Science & ICT (NRF-2012M3A9C4048761); The funders had no role in study design, data collection and analysis, decision to publish, or preparation of the manuscript. This work was also supported by Sempio Co., Ltd (Seoul, South Korea); The funders had no role in study design, data collection and analysis, decision to publish, or preparation of the manuscript. There was no additional external funding received for this study.

**Competing interests:** The authors have declared that no competing interests exist.

## Introduction

Normal functioning of the immune system is essential for physical health, and dietary compounds are believed to play a role in maintaining or strengthening immune function [1]. To demonstrate beneficial immune effects of food ingredients, food regulatory authorities in general request proof of reduced incidence, severity, or duration of infections. Additionally, some regulatory authorities allow the use of vaccination-induced immune responses to demonstrate the effect of beneficial physiological effects of food ingredients or supplements on the immune system [2].

In the general population, increased inflammation or immune responses due to challenge could provide more sensitive indications than measuring markers in steady-state, and vaccination could potentially be a standardized challenge applicable to nutritional intervention studies [3]. Dietary modulation of the response to oral vaccination could be a good model for studying the effect on gastrointestinal inflammation and gut permeability. The Attenuated enterotoxigenic *Escherichia coli* (ETEC) strain, developed initially as a potential vaccine, is already used in a challenge model. However, as it induces gastrointestinal symptoms, the primary readout in this model is the symptoms [4, 5]. To investigate gastrointestinal inflammation and immune response, a model with a lower burden for study participants is preferred.

The oral cholera vaccine (Dukoral®) consists of $1.25 \times 10^{11}$ inactivated Vibrio cholera bacteria and 1 mg recombinant cholera toxin B-subunit (rCTB) per dose, provided with Sodium Hydrogen Carbonate. CTB is non-toxic and is a potent mucosal immunogen due to its affinity binding to the receptor GM1 ganglioside [6]. It could be used as a carrier for antigens and improve mucosal immunity [7]. It is hypothesized that the binding of the CTB induces an inflammatory reaction, however how strong or how extensive the inflammatory reaction is, has not been investigated yet.

*Lactobacillus plantarum* is listed in the Korean Health Functional Food code [8] and is a representative probiotic that appears in the fermenting season among the lactic acid bacteria in Kimchi, a Korean traditional fermented food. Lipoteichoic acid (LTA), a component acquired from the cell wall, was considered a proinflammatory molecule of Gram-positive bacteria [9]. However, in several studies, LTA separated from beneficial bacteria such as *Lb. plantarum* showed weak nitric oxide production compared to pathogen and attenuated the induced inflammation [9–12]. Also, our previous study showed that *Lb. plantarum* K8 lysate regulated Toll like receptor-2 signaling, prevented DSS-induced intestinal mucosal damage, and reduced increased proinflammatory cytokines in rats [13]. Therefore, we expect *Lb. plantarum* K8 lysate to help regulate the balance of the intestinal immune system. However, the efficacy of *Lb. plantarum* K8 lysate on gut immunity in the general population has rarely been studied.

In this study, we aimed to develop a new human intervention study protocol that can be used to study gastrointestinal inflammation. Because of the immunogenicity of the cholera vaccine targeting the gut, we aim to provide support for our concept that, in addition to inducing a protective immune response, this vaccine might also be used as a safe approach to induce a mild gastrointestinal inflammatory response in healthy human subjects, as a challenge model for dietary intervention studies. To study whether this inflammatory response or the vaccination response can be modulated by an active ingredient, we use *Lb. plantarum* K8 lysate as an intervention.

## Methods

### Ethics approval

The Institutional Review Board (IRB) of Ewha Womans University Medical Center approved the study (EUMC 2017-05-068) on 05/07/17. This trial is registered in the International

| | STUDY PERIOD | | | | | | | | | | | | | | | |
| | Enrollment (day) | Allocation (day) | Treatment (day) | | | | | | | | | | | | | |
| TIMEPOINT | 0 | 0 | 0 | 1 | 14 | 15 | 16 | 17 | 19~27 | 28 | 29 | 30 | 31 | 32~41 | 42 | 43 |
| Visit | 1 | 1 | 1 | | | 2 | 3 | 4 | | | 5 | | 6 | | | 7 |
| **ENROLMENT:** | | | | | | | | | | | | | | | | |
| Eligibility screen | √ | | | | | | | | | | | | | | | |
| Informed consent | √ | | | | | | | | | | | | | | | |
| Allocation | | √ | | | | | | | | | | | | | | |
| **INTERVENTIONS:** | | | | | | | | | | | | | | | | |
| Probiotic lysate | | | | √ | √ | √ | √ | √ | √ | √ | √ | √ | √ | √ | √ | √ |
| Placebo | | | | √ | √ | √ | √ | √ | √ | √ | √ | √ | √ | √ | √ | √ |
| Vaccination | | | | | | √ | | | | | √ | | | | | |
| **ASSESSMENTS:** | | | | | | | | | | | | | | | | |
| _Effective variables_ | | | | | | | | | | | | | | | | |
| Calprotectin (Feces) | | | √ | | | √ | √ | √ | | | √ | √ | √ | | | √ |
| Cholera toxin specific antibodies (IgA, IgG, plasma) | | | | | | √ | | | | | √ | | | | | √ |
| Beta defensin (Feces) | | | √ | | | √ | √ | √ | | | √ | √ | √ | | | √ |
| I-FABP (Plasma) | | | | | | √ | √ | √ | | | √ | | √ | | | √ |
| Blood cellular markers | | | √ | | | √ | √ | √ | | | √ | | √ | | | √ |
| IP10 (Plasma) | | | | | | √ | √ | √ | | | √ | | √ | | | √ |
| Acute-phase proteins (hsCRP, IL-1ra, Plasma) | | | | | | √ | √ | √ | | | √ | | √ | | | √ |
| Microbiome (Feces) | | | √ | | | √ | | | | | | | | | | √ |
| Bristol stool scale questionnaire | | | √ | √ | √ | √ | √ | √ | √ | √ | √ | √ | √ | √ | √ | √ |
| _Safety variables_ | | | | | | | | | | | | | | | | |
| Blood chemistry (CBC) | | | √ | | | √ | √ | √ | | | √ | | √ | | | √ |
| AST, ALT (Serum) | | | √ | | | | | | | | | | | | | √ |
| Vital signs | | | √ | | | √ | √ | √ | | | √ | | √ | | | √ |
| Adverse events | | | √ | √ | √ | √ | √ | √ | √ | √ | √ | √ | √ | √ | √ | √ |

**Fig 1. SPIRIT flow diagram.** Flowchart of schedule of enrollments, interventions, and assessments. IgA = immunoglobulin A, IgG = immunoglobulin G, I-FABP = Intestinal fatty-acid binding protein, IP10 = Interferon gamma-induced protein 10, hsCRP = high-sensitivity C-reactive protein, IL-1ra = interleukin-1 receptor antagonist, AST = aspartate aminotransferase, ALT = alanine aminotransferase.

Clinical Trials Registry Platform of WHO (ICTRP, registration number: KCT0002589) and available at https://cris.nih.go.kr. This study will be performed according to Good Clinical Practice and the Declaration of Helsinki. Informed consent will be documented with a written consent form approved by the IRB and signed by the participant or the participant's legally authorized representative. A copy will be given to the person who signed the form.

## Study design

This study is a randomized, double-blind, placebo-controlled, parallel-group study investigating the efficacy and safety of probiotics lysate in modulating local and systemic inflammation. Participants will be screened according to inclusion/exclusion criteria, including a brief medical examination after voluntarily signing the informed consent form. Eligible subjects will be randomly allocated to either treatment or placebo groups. The schedule of enrollments, interventions, and assessments is shown in the SPIRIT flow diagram (Fig 1).

## Sample size calculation

The sample size for the placebo group and the treatment group will be registered for a total of 46 people, 23 per group, in consideration of the 20% drop-out rate. The required sample size was calculated for the calprotectin level in feces and C-reactive protein (CRP) in plasma.

According to the results of previous research, we assumed that the calprotectin level (mean ± SD) in healthy individuals is 15 ± 20 μg/g feces [4]. In healthy adults, an increase to 50 μg/g is considered an upper limit [14]. Based on two-sided statistical testing for paired data, α = 0.05 (chance on a type-I error) and β = 0.10 (chance on a type-II error), 19 subjects per group are needed for this outcome. For C-reactive protein, it was shown that the plasma level in low-grade inflammation is 2.54 ± 2.07 mg/L (mean ± SD), as compared to 1.01 ± 0.88 mg/L in normal individuals [15]. Assuming that cholera vaccination will increase CRP to a similar extent compared to baseline in the placebo group, based on two-sided statistical testing for paired data, α = 0.05 (chance on a type-I error) and β = 0.20 (chance on a type-II error), 18 subjects are calculated per group for this outcome.

## Participants and eligibility criteria

Participants will include males between 20 and 50 years old. Considering the dropout rate of 20%, 46 eligible participants will be recruited from Ewha Womans University Medical Center. The inclusion/exclusion criteria are as follows.

Inclusion Criteria:

1. A subject who voluntarily agrees to participate and sign the informed consent form

2. Male (aged 20 to 50 years)

3. Healthy bowel habits (A person who defecates regularly at least four times a week and once a day (less than two times))

4. Availability of internet connection

5. Smartphone user

Exclusion criteria:

1. Previous cholera vaccination history

2. Other vaccination in the past 1 month

3. Acute gastroenteritis in the past 2 months

4. Use of antibiotics in the past 2 months

5. Hypersensitivity to the vaccine, to formaldehyde, to any of the excipients (sodium salts), or to probiotics

6. The disease of GI tract (constipation, diarrhea, Inflammatory bowel disease, etc), liver, gall bladder, kidneys, thyroid gland

7. Immune-compromised

8. Use of immunosuppressive drugs

9. Drug abuse, and not willing/able to stop this during the study

10. Continuous consumption of probiotics within 2 weeks prior to the study

11. Excessive alcohol usage (140 g/week, about 3.5 bottles/week or 4 glasses/day as soju)

12. Participating in another clinical trial within 12 weeks prior to or during the study

13. Visiting a country affected by cholera in the past 1 month

### Recruitment

Subjects will recruit using posters and advertisements (online and/or offline) with a brief explanation about the research and the investigator's contact information in Ewha Womans University Medical Center and the surrounding area. Those who want to participate in this study will obtain detailed information about the study from trained research coordinators.

### Randomization, allocation concealment, and binding

Participants will be randomly assigned (1:1) to a placebo or treatment group. Study products will be provided by Sempio Co., Ltd. (Seoul, Korea). This will be a double-blind trial, the product will be blinded to the subjects, investigators, and coordinators throughout the study period. The code list identifying the product consumed by each subject will be opened only at the end of the study, after data analysis, or earlier in case of medical necessity.

### Handling of withdrawal

Participants can withdraw from the study for any reason at any time. If the subject decides to withdraw the consent, the investigator will ask the reason for the withdrawal and document it in the case report form (CRF). The subject may be removed from the study for the following reason:

1. A subject requests discontinuation; the principal investigator (PI) initiates removal for medical or compliance reasons

2. Occurrence of an adverse event that is considered serious by the PI

3. Occurrence of an illness that affects the subject's further participation

4. Any other reason considered by the PI to be necessary for a subject's safety

5. Non-compliance with study procedures

### Interventions

All subjects will receive the registered oral cholera vaccine Dukoral® according to the standard vaccination scheme on visit 2 and visit 5 (2 doses, 14 days apart). Participants will take probiotic lysate material or a placebo for 6 weeks (day 1–43). Both products will be coated with identical colors and shapes and will be provided to the subjects in bottles. A daily serving consists of two chewable tablets, to be taken as one tablet twice a day.

### Outcome measures

Outcomes related to efficacy and safety will be evaluated. Trained investigators and research coordinators will conduct interviews and blood sampling of subjects. Participants will be instructed to visit after a 12-hour overnight fast on each visit. The vaccination time points are fixed on day 15 and day 29. On the day of taking the vaccine, blood must be collected before providing the vaccine. Fecal samples will be collected on days 0, 14, 16, 17, 28, 30, 31, and 42 at the subjects' homes. Fecal sampling will be allowed on day 1 or day 43, respectively, when it is difficult to collect samples on day 0 or day 42. If subjects cannot collect a fecal sample on day 0 or day 1, they will collect a fecal sample as soon as possible and start intake of the study product the day after collecting the first fecal sample. The stool sample is frozen in the home freezer and can be submitted with ice packs the next day or delivered to the laboratory within an hour through the delivery system provided.

## Efficacy variables

To assess local inflammatory response, calprotectin and beta-defensin levels will be analyzed in feces at visits 1–7. At visits 2, 5, and 7, cholera toxin-specific Immunoglobulin A (IgA) and Immunoglobulin G (IgG) levels in plasma will be confirmed. Intestinal fatty-acid binding protein (I-FABP) as an intestinal epithelial damage/repair marker, cytokines/chemokines (Interferon gamma-induced protein 10 (IP-10), etc.), acute-phase proteins such as high-sensitivity CRP and Interleukin 1 receptor antagonist (IL-1ra) in plasma will be analyzed to further explore the systemic inflammatory response on visit 2–7. Blood cellular markers (leukocytes, lymphocytes, neutrophils, etc.) will be monitored at every visit. Analyses for the gut microbiome will be performed on visits 1, 2, and 7. Bristol stool scale questionnaire will be conducted by online survey every day.

## Safety variables

Blood chemistry and vital signs will be monitored at every visit. Aspartate Aminotransferase (AST) and Alanine Aminotransferase (ALT) will be measured on days 0 and 43. Vital signs (body temperature, systolic/diastolic blood pressure, and pulse rate) will be confirmed at every visit. Unusual signs will be monitored by online questionnaires during the study. Unintended changes in pathological, anatomical, metabolic, or physiological functions associated with the use of a study product are considered side effects, except for changes associated with normal activities that are ordinarily clinically expected to change in frequency or magnitude form. The PI will assess all AEs as to their severity and relation to the study product. Appropriate treatment and follow-up will be performed, which will continue until the condition is resolved or the cause is identified.

## Possible covariables

Age, medical history, and lifestyle (alcohol history and smoking use) will be assessed at baseline. Participants' current medication history and supplement use will be checked at each visit. Weight will be measured at each visit, and height will be measured on the first visit. Body mass index (BMI) will be calculated using measured weight and height [weight (kg)/height (m$^2$)]. The same researcher will use the same equipment to measure parameters for each visit. Subjects will also fill out our recommended food score questionnaire at the beginning and submit their diet diary every week, 2 days on weekdays, and 1 day on weekends using a mobile application during the study for monitoring dietary compliance. Participants will be asked to limit intakes of health functional foods, avoid the consumption of pre-or probiotics products such as dietary fiber, oligosaccharide, etc. In addition, alcohol consumption will not be permitted as well as the day before sampling (days -1 and 0, days 13–17, days 27–31, days 41–43).

## Compliance

Compliance will be assessed biweekly by interviewing subjects and counting the study product returned to the clinic. Consuming less than 80% of the scheduled intakes will be defined as non-compliance. Subjects will be called as appropriate to encourage compliance.

## Data management and quality assurance

All data generated by the methods described in the study protocol will be recorded on the electronic CRFs. Steps taken to assure the reliability and accuracy of the study data include the selection of qualified PIs, research coordinators, research assistants, and contract laboratory; review of protocol procedures with the PIs, protocol writer, and associated personnel before

the study; and the presence of a research coordinator during the study days. The research coordinator will review CRFs for accuracy and completeness. The sponsor will carry out periodic on-site monitoring of study procedures and documents.

## Statistical analysis

Data analyses will be performed with intention-to-treat subject data (all data from subjects who have received the study product) or per-protocol subject data (only data from subjects who completed the study at least 80% compliant with the study product). Variables will be tested for normal distribution and data transformation will be performed on skewed variables. Parametric or non-parametric analyses (dependent on the outcome of the distribution of data) such as paired *t*-test, Student's *t*-test, or Wilcoxon signed-rank test will be used to analyze the difference within each group or between the groups for continuous variables. Chi-square or Fisher's exact test will be used to analyze the difference between the groups for categorical variables. A linear mixed-effects model with group and time as fixed effects and subject as the random effect will be applied to compare the changes between groups over the intervention period. The correlation between variables will be analyzed by Pearson's correlation or Spearman correlation and regression analysis will be performed. The data will be analyzed using the SAS 9.4 statistical software. The difference between the groups will be tested two-sided for all study outcomes. *P*-values $<0.05$ are classified as statistically significant.

## Discussion

This is a randomized, double-blind study that evaluates the effectiveness of dietary intervention against gut stimulation induced by the oral cholera vaccine. A new model of gut stimulation is required to evaluate changes in intestinal immune function due to dietary modulation. Since the intestinal immune response cannot be determined with a single marker [1], the assessment of several immune markers is included. In previous human studies with the oral cholera vaccine, the immune response was mainly studied by analyzing the vaccine-specific antibody response [16]. To evaluate the efficacy of dietary interventions in this gut stimulation model, we will further analyze markers related to intestinal inflammation, microbial translocation, and systemic immune activation.

To assess the effect of dietary modulation in this cholera vaccine model, we will use *Lb. plantarum* lysate. Probiotics, a living microorganisms, are known to provide health benefits to the host when administered in an appropriate dosage [17]. According to recent studies, it has been reported that lysate acquired from probiotics has similar effects to probiotics [18–20] and administration of killed bacteria or lysate of the *Lb. plantarum* genus also regulated cytokine production [13, 21].

The regulation of immune responses differs due to the presence of different hormones in males and females. The menstrual cycle represents hormonal fluctuations for immune function and regulation. In particular, the number of immune cells and regulated activity during the four-week cycle, as demonstrated in the case of regulatory T cells [22, 23]. Recently, several studies have demonstrated relevant gender differences in microbiota composition that can account for differences in peripheral immunity between genders as well as intestinal immunity. In the research conducted on rodents, female mice showed increased microbial diversity compared to male mice [24–27]. Hence, the reproductive condition of females that may be related to microbiota results as confounder factors should be considered, and participants in this protocol consisted only of males.

In this study, we describe clinical trials which will evaluate the effect of *Lb. plantarum* lysate on the inflammatory response or on the immune response, suggesting the protocol that the

oral cholera vaccination can be used as a model to induce a mild inflammatory response in the gut. Oral cholera vaccinations are used to induce an immune response in the intestine to prevent cholera infection. However, oral vaccines can also affect the immune response of other mucosal tissues. Oral vaccination primarily interacts with the immune system through the oral cavity to the small intestine and induces the expression of vaccine-specific IgA in gastrointestinal and tissue-specific homing of B cells and T cells [28]. The aim of the measurement of cholera toxin-specific IgA was to evaluate whether oral cholera vaccination is able to induce potent IgA responses in serum.

We hypothesized first, the oral cholera vaccination will induce a significant increase in the gut inflammatory response in the gastrointestinal tract, and second, *Lb. plantarum* K8 lysate will significantly improve the inflammatory response or support the immune response induced by oral cholera vaccination in healthy male subjects. To our knowledge, this is the first randomized, double-blind study to investigate the efficacy of *Lb. plantarum* lysate on the inflammatory response as induced by oral cholera vaccination. We expect that the results of this study will contribute to the scientific evidence that the tested probiotic lysate has the potential of modulating intestinal inflammation induced by the oral cholera vaccine or improving the immune response.

## Supporting information

**S1 File. SPIRIT checklist.**
(PDF)

**S2 File. Protocol.**
(PDF)

## Acknowledgments

The authors thank Liz Kamei of Nizo for enabling Ewha-Nizo collaboration.
    The authors have no conflicts of interest to disclose.

## Author Contributions

**Conceptualization:** Els van Hoffen, Oran Kwon, Ji Yeon Kim.

**Data curation:** Min Young Park.

**Formal analysis:** Min Young Park.

**Funding acquisition:** Oran Kwon.

**Investigation:** Min Young Park, Joohee Kim, Hee Jung Choi.

**Methodology:** Min Young Park, Els van Hoffen.

**Project administration:** Anita Hartog, Alwine Kardinaal, Hee Jung Choi, Oran Kwon, Ji Yeon Kim.

**Supervision:** Oran Kwon, Ji Yeon Kim.

**Validation:** Joohee Kim.

**Writing – original draft:** Min Young Park, Soo-yeon Park, Anita Hartog, Ji Yeon Kim.

**Writing – review & editing:** Soo-yeon Park, Alwine Kardinaal.

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
