## [Decision Letter · Decision Letter 0]

21 Sep 2022

PONE-D-22-04882Study protocol for cholera vaccination as a model to measure the inflammatory response in the gut: A case of modulation with a Lactobacillus plantarum K8 lysatePLOS ONE

Dear Dr. Kim,

Thank you for submitting your manuscript to PLOS ONE. After careful consideration, we feel that it has merit but does not fully meet PLOS ONE’s publication criteria as it currently stands. Therefore, we invite you to submit a revised version of the manuscript that addresses the points raised during the review process. Your manuscript has been assessed by one peer-reviewer and their report is appended below. The reviewer comments that certain aspects of the study require further detail, clarification, or justification.  Could you please revise the manuscript to carefully address the concerns raised? Please note that we have only been able to secure a single reviewer to assess your manuscript. We are issuing a decision on your manuscript at this point to prevent further delays in the evaluation of your manuscript. Please be aware that the editor who handles your revised manuscript might find it necessary to invite additional reviewers to assess this work once the revised manuscript is submitted. However, we will aim to proceed on the basis of this single review if possible. 

We look forward to receiving your revised manuscript.

Kind regards,

Maria Elisabeth Johanna Zalm, Ph.D

Editorial Office

PLOS ONE

Journal Requirements:

“This work was primarily supported by the "Bio&Medical Technology Development Program" of the Natl. Research Foundation (NRF) funded by the Ministry of Science & ICT (NRF-2012M3A9C4048761); The funders had no role in study design, data collection and analysis, decision to publish, or preparation of the manuscript.

This work was also supported by Sempio Co., Ltd (Seoul, South Korea); The funders had no role in study design, data collection and analysis, decision to publish, or preparation of the manuscript.”

“This work was primarily supported by the "Bio&Medical Technology Development Program" of the Natl. Research Foundation (NRF) funded by the Ministry of Science & ICT (NRF-2012M3A9C4048761).

This work was also supported by Sempio Co., Ltd (Seoul, South Korea).

The authors thank Liz Kamei of Nizo for enabling Ewha-Nizo collaboration.

The authors have no conflicts of interest to disclose.”

“This work was primarily supported by the "Bio&Medical Technology Development Program" of the Natl. Research Foundation (NRF) funded by the Ministry of Science & ICT (NRF-2012M3A9C4048761); The funders had no role in study design, data collection and analysis, decision to publish, or preparation of the manuscript.

This work was also supported by Sempio Co., Ltd (Seoul, South Korea); The funders had no role in study design, data collection and analysis, decision to publish, or preparation of the manuscript.”

Reviewers' comments:

Reviewer's Responses to Questions

**Comments to the Author**

1. Does the manuscript provide a valid rationale for the proposed study, with clearly identified and justified research questions?

Reviewer #1: Yes

2. Is the protocol technically sound and planned in a manner that will lead to a meaningful outcome and allow testing the stated hypotheses?

Reviewer #1: Yes

3. Is the methodology feasible and described in sufficient detail to allow the work to be replicable?

Reviewer #1: Yes

4. Have the authors described where all data underlying the findings will be made available when the study is complete?

Reviewer #1: Yes

5. Is the manuscript presented in an intelligible fashion and written in standard English?

Reviewer #1: Yes

6. Review Comments to the Author

You may also provide optional suggestions and comments to authors that they might find helpful in planning their study.

Reviewer #1: The study aimed to evaluate the intestinal stimulation induced by the oral cholera vaccine in healthy people.

1) Why was the study population made up of only male individuals?

2) What did the authors consider healthy bowel habits?

3) Why was the 6 week supplementation time with probiotics chosen? What is the justification for this period?

4) Why did the authors not assess fecal secretory IgA?

5) What about serum LPS and zonulin levels?

6) Evaluating the variations in the intestinal microbiota would be of great value to the study.

7) Improve the conclusions? What is the clinical significance of the study? What diseases can be studied with this model?

7. PLOS authors have the option to publish the peer review history of their article (what does this mean?). If published, this will include your full peer review and any attached files.

Reviewer #1: **Yes: **Gislane Lelis Vilela de Oliveira

---

## [Author Response · Author response to Decision Letter 0]

31 Oct 2022

Dear Editor,

Thank you very much for your consideration of our manuscript and request for a revised version. We have copied and pasted all reviewers’ comments and addressed each one individually. As you will see, we made a number of changes in our manuscript to incorporate the questions and suggestions by the reviewers as thoroughly as possible. Below is a list of the changes following the order of the comments.

Reviewer #1

1. Why was the study population made up of only male individuals?

-> We thank the reviewer for this important comment.

The regulation of immune responses differs due to the presence of different hormones in males and females. The menstrual cycle represents hormonal fluctuations for immune function and regulation. In particular, the number of immune cells and regulated activity during the four-week cycle, as demonstrated in the case of regulatory T cells. 

Recently, several studies have demonstrated relevant gender differences in microbiota composition that can account for differences in peripheral immunity between genders as well as intestinal immunity. In the research conducted on rodents, female mice showed increased microbial diversity compared to male mice.

Hence, the reproductive condition of females that may be related to microbiota results as confounder factors should be considered. This is the reason for the participants of this study being made up of only males (Lines 270-278).

2. What did the authors consider healthy bowel habits?

-> We conducted a questionnaire on bowel habits. This study confirmed healthy bowel habits as a person who defecates regularly at least four times a week and once a day (less than two times). 

We added in Lines 133-134.

3. Why was the 6 week supplementation time with probiotics chosen? What is the justification for this period?

-> It was intended to improve immunity by taking probiotics for 2 weeks before the first vaccination, taking them for 2 weeks between the first vaccination and the second vaccination, and taking them for 2 weeks after the second vaccination to demonstrate the effect of the probiotics with vaccine administration.

4. Why did the authors not assess fecal secretory IgA?

->Oral cholera vaccinations are used to induce an immune response in the intestine to prevent cholera infection. However, oral vaccines can also affect the immune response of other mucosal tissues. Oral vaccination primarily interacts with the immune system through the oral cavity to the small intestine and induces the expression of vaccine-specific IgA in gastrointestinal and tissue-specific homing of B cells and T cells. The aim of the measurement of cholera toxin-specific IgA was to evaluate whether oral cholera vaccination is able to induce potent IgA responses in serum. We added Lines 283-289.

5. What about serum LPS and zonulin levels?

->Fecal calprotectin level, β-defensin, I-FABP (Intestinal fatty acid-binding protein), and other cytokines/chemokines were biomarkers of this study protocol. This is important to understand the kinetics and the persistence of the effect of the vaccination on these markers.

6. Evaluating the variations in the intestinal microbiota would be of great value to the study.

->We thank the reviewer for this important comment. We are planning to measure the microbiome after breaking the blind. The study is ongoing. We also have included the sentences in Lines 263-265.

7. Improve the conclusions? What is the clinical significance of the study? What diseases can be studied with this model?

->This paper is for clinical protocol. Clinical analysis is currently in progress. We will analyze and publish the paper afterward.

---

## [Decision Letter · Decision Letter 1]

2 Feb 2023

Study protocol for cholera vaccination as a model to measure the inflammatory response in the gut: A case of modulation with a Lactobacillus plantarum K8 lysate

PONE-D-22-04882R1

Dear Dr. Kim,

We’re pleased to inform you that your manuscript has been judged scientifically suitable for publication and will be formally accepted for publication once it meets all outstanding technical requirements.

Kind regards,

Brenda A Wilson, Ph.D.

Academic Editor

PLOS ONE

Additional Editor Comments (optional):

Reviewers' comments:

Reviewer's Responses to Questions

**Comments to the Author**

1. Does the manuscript provide a valid rationale for the proposed study, with clearly identified and justified research questions?

Reviewer #1: Yes

Reviewer #2: Yes

2. Is the protocol technically sound and planned in a manner that will lead to a meaningful outcome and allow testing the stated hypotheses?

Reviewer #1: Yes

Reviewer #2: Yes

3. Is the methodology feasible and described in sufficient detail to allow the work to be replicable?

Reviewer #1: Yes

Reviewer #2: Yes

4. Have the authors described where all data underlying the findings will be made available when the study is complete?

Reviewer #1: Yes

Reviewer #2: Yes

5. Is the manuscript presented in an intelligible fashion and written in standard English?

Reviewer #1: Yes

Reviewer #2: Yes

6. Review Comments to the Author

You may also provide optional suggestions and comments to authors that they might find helpful in planning their study.

Reviewer #1: The manuscripts aims to evaluate the gut stimulation induced by oral cholera vaccine in healthy individuals and the efficacy and safety of a probiotic lysate, identifying whether functional ingredients in food can modulate inflammatory response induced by oral cholera vaccine.

The authors revised the manuscript and answered all my questions.

Reviewer #2: This is a revised version of a previously submiited manuscript. Here, the authors were able to successfully address all previous questions. I have no further questions/comments.

7. PLOS authors have the option to publish the peer review history of their article (what does this mean?). If published, this will include your full peer review and any attached files.

Reviewer #1: **Yes: **Gislane Lelis Vilela de Oliveira

Reviewer #2: No

---

## [Editor Report · Acceptance letter]

9 Feb 2023

PONE-D-22-04882R1 

Study protocol for cholera vaccination as a model to measure the inflammatory response in the gut: A case of modulation with a *Lactobacillus plantarum* K8 lysate 

Dear Dr. Kim:

I'm pleased to inform you that your manuscript has been deemed suitable for publication in PLOS ONE. Congratulations! Your manuscript is now with our production department. 

Kind regards, 

on behalf of

Dr. Brenda A Wilson 

Academic Editor

PLOS ONE